# Assessing the efficacy, safety and utility of closed-loop insulin delivery compared with sensor-augmented pump therapy in very young children with type 1 diabetes (KidsAP02 study): an open-label, multicentre, multinational, randomised cross-over study protocol

Julia Fuchs [1,2] Janet M Allen,[1] Charlotte K Boughton [1]
Malgorzata E Wilinska,[1] Ajay Thankamony,[2] Carine de Beaufort,[3] Fiona Campbell,[4] James Yong,[4] Elke Froehlich-Reiterer,[5] Julia K Mader,[6] Sabine E Hofer,[7] Thomas M Kapellen,[8] Birgit Rami-Merhar,[9] Martin Tauschmann,[9] Korey Hood,[10] Barbara Kimbell,[11] Julia Lawton,[11] Stephane Roze,[12] Judy Sibayan,[13] Nathan Cohen,[13] Roman Hovorka,[1,2] on behalf of the KidsAP Consortium

► Prepublication history and supplemental material for this paper is available online. To view these files, please visit the journal online (http://dx.doi.org/10.1136/bmjopen-2020-042790).

For numbered affiliations see end of article.

**Correspondence to**
Professor Roman Hovorka;
rh347@cam.ac.uk

## ABSTRACT

**Introduction** Diabetes management in very young children remains challenging. Glycaemic targets are achieved at the expense of high parental diabetes management burden and frequent hypoglycaemia, impacting quality of life for the whole family. Our objective is to assess whether automated insulin delivery can improve glycaemic control and alleviate the burden of diabetes management in this particular age group.

**Methods and analysis** The study adopts an open-label, multinational, multicentre, randomised, crossover design and aims to randomise 72 children aged 1–7 years with type 1 diabetes on insulin pump therapy. Following screening, participants will receive training on study insulin pump and study continuous glucose monitoring devices. Participants will be randomised to 16-week use of the hybrid closed-loop system (intervention period) or to 16-week use of sensor-augmented pump therapy (control period) with 1–4 weeks washout period before crossing over to the other arm. The order of the two study periods will be random. The primary endpoint is the between-group difference in time spent in the target glucose range from 3.9 to 10.0 mmol/L based on sensor glucose readings during the 16-week study periods. Analyses will be conducted on an intention-to-treat basis. Key secondary endpoints are between group differences in time spent above and below target glucose range, glycated haemoglobin and average sensor glucose. Participants' and caregivers' experiences will be evaluated using questionnaires and qualitative interviews, and sleep quality will be assessed. A health economic analysis will be performed.

**Ethics and dissemination** Ethics approval has been obtained from Cambridge East Research Ethics Committee (UK), Ethics Committees of the University of Innsbruck, the University of Vienna and the University of Graz (Austria),

### Strengths and limitations of this study

► The study adopts an open-label, multinational, multicentre, randomised, crossover design, and includes a large group of children across multiple geographical locations.
► The study enrols very young children with type 1 diabetes and assesses closed-loop insulin delivery over 4 months in the home setting.
► All participants are already pump users limiting generalisability for those on multiple daily injections.
► The comparator group uses sensor-augmented pump therapy without low glucose suspend.
► The study includes psychosocial assessments including sleep quality and health economic analysis to support adoption of closed-loop systems in this population.

Ethics Committee of the Medical Faculty of the University of Leipzig (Germany) and Comité National d'Ethique de Recherche (Luxembourg). The results will be disseminated by peer-reviewed publications and conference presentations.
**Trial registration number** NCT03784027.

## INTRODUCTION

Type 1 diabetes (T1D) is characterised by a deficiency of insulin and is caused by immune-mediated destruction of pancreatic beta-cells in genetically predisposed individuals.[1] The incidence of T1D is increasing by approximately 3% annually,[2] including in the

youngest age group.[3] Achieving tight glycaemic control in T1D is particularly challenging in this population due to high insulin sensitivity, unpredictable food intake and physical activity, and highly variable insulin requirements.[4 5] Hypoglycaemia is frequently asymptomatic and prolonged, particularly at night-time.[6–9] Caregivers' fear of hypoglycaemia leads to high diabetes management burden, sub-optimal glycaemic control and reduced quality of life.[10] This contributes to the majority of young children either failing to meet treatment guidelines for target glycated haemoglobin (HbA1c) below 7.0% (53mmol/mol),[11–14] or only achieving good glycaemic control through frequent caregiver input leading to high management burden.[15 16] Poorer glycaemic control and young age at diagnosis of T1D are associated with structural white and grey matter changes and significant differences in brain growth compared with healthy controls.[17 18] Frequent hyperglycaemia and longer duration of T1D increase the risk of late microvascular and macrovascular complications.[19]

Over the past decades, new technologies have improved management of T1D. The use of insulin pumps is increasing, with 60%–92% of those below the age of 6 years using pumps.[20 21] Real-time continuous glucose monitoring (CGM) is now widely used, providing alarms and facilitating responsive insulin dose adjustments.[22] CGM use is associated with reduced incidence of diabetic ketoacidosis and severe hypoglycaemia, but improvements in HbA1c are modest.[23 24] Despite increased technology use, optimal glycaemic control in preschool children remains challenging.

Automated insulin delivery systems are an emerging technology promising to transform management of T1D.[25 26] Systems use real-time glucose monitoring to inform algorithm-directed insulin delivery via an insulin pump. This aims to achieve near-physiological glucose-responsive subcutaneous insulin delivery.[27] Current hybrid systems continue to require user-initiated prandial insulin boluses. Several hybrid closed-loop systems have been evaluated in children and adolescents. Results demonstrate improved glycaemic control, reduced risk of hypoglycaemia and decreased diabetes management burden with improvement in quality of life for children and caregivers.[28–35] Four hybrid closed-loop systems are now commercially available: the 670G and 780G systems (Medtronic, Northridge, California, USA), the Control-IQ system (Tandem Diabetes Care, San Diego, California, USA) and the CamAPS FX app (CamDiab, Cambridge, UK). Evaluations of hybrid closed-loop systems in preschool children in the home setting have been limited to 3 weeks' duration.[34] Longer-term studies are required to assess the efficacy of closed-loop therapy in this particular population.

We hypothesise that the closed-loop approach using the Cambridge closed-loop algorithm implemented in the CamAPS FX app (figure 1) will increase time in target glucose range over 16 weeks as compared with sensor-augmented pump therapy in very young children

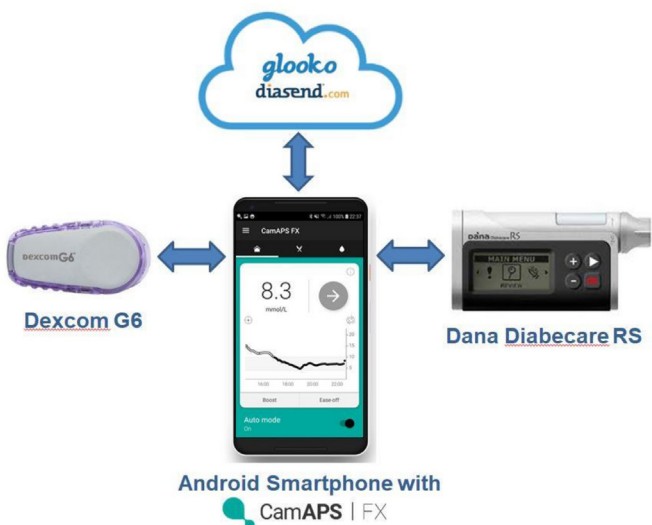

**Figure 1** CamAPS FX app with Dexcom G6 sensor (Dexcom, San Diego, California, USA), Dana Diabecare RS insulin pump (Sooil Development, Seoul, Korea), and automatic upload capabilities to diabetes management system Diasend (Glooko/Diasend, Göteborg, Sweden) (reproduced with permission from CamDiab, Cambridge, UK).

with T1D. We will assess safety and acceptability of this therapy, evaluate its impact on quality of life for caregivers, and perform cost–utility analysis to inform potential reimbursement.

## METHODS AND ANALYSIS
### Overview
The study adopts an open-label, multicentre, multinational, randomised, two-period crossover design contrasting closed-loop insulin delivery using the Cambridge closed-loop algorithm in very young children with T1D with sensor-augmented pump therapy (figure 2). Participants will include children aged 1–7 years on insulin pump therapy for a minimum of 3 months. The study aims to randomise 72 participants, with each centre recruiting 8–12 participants. The two intervention periods will last 16 weeks each, the order of which will be random. There will be seven clinical sites across the EU (KidsAP Horizon 2020 project).

The University of Cambridge (UK) will be the coordinating centre. Clinical sites include:

1. Addenbrooke's Hospital, Cambridge University Hospital NHS Foundation Trust, Cambridge, UK.
2. Leeds Teaching Hospitals NHS Trust, Leeds, UK.
3. DECCP, Centre Hospitalier de Luxembourg, Grand Duché de Luxembourg.
4. Hospital for Children and Adolescents, University of Leipzig, Leipzig, Germany.
5. Department of Pediatric and Adolescent Medicine, Medical University of Graz, Graz, Austria.
6. Department of Pediatrics I, Medical University of Innsbruck, Innsbruck, Austria.

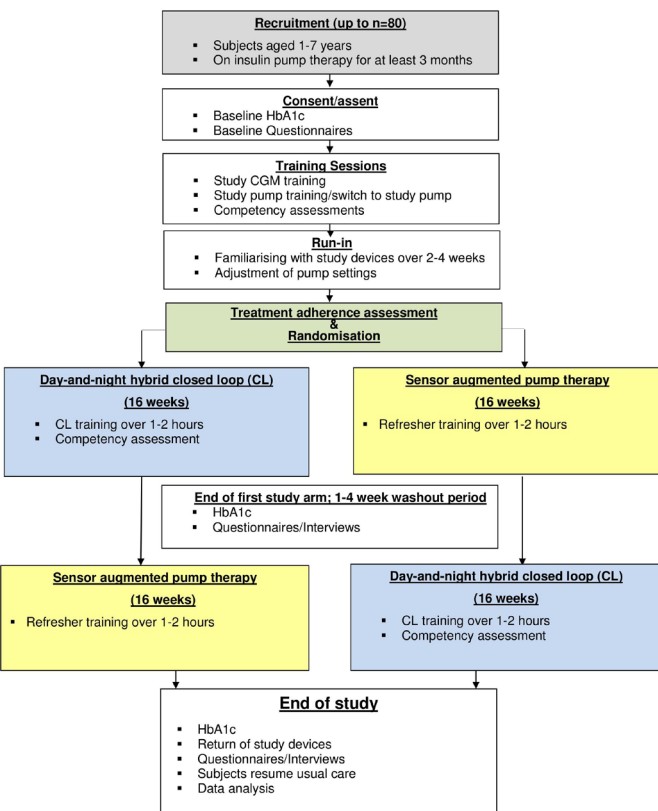

**Figure 2** Study flowchart. CGM, continuous glucose monitoring; HbA1c, glycated haemoglobin.

7. Department of Pediatrics and Adolescent Medicine, Medical University of Vienna, Vienna, Austria.

Participants may be recruited from Patient Identification Centres associated with these sites. Qualitative interviews with parents/guardians will be carried out by the University of Edinburgh (Edinburgh, UK). Health economics evaluation will be conducted by VYOO Agency (Lyon, France).

### Inclusion criteria
1. Age between 1 and 7 years (inclusive).
2. T1D (WHO definition: 'The aetiological type named type 1 encompasses the majority of cases which are primarily due to beta-cell destruction, and are prone to ketoacidosis. Type 1 includes those cases attributable to an autoimmune process, as well as those with beta-cell destruction for which neither an aetiology nor a pathogenesis is known (idiopathic). It does not include those forms of beta-cell destruction or failure to which specific causes can be assigned (eg, cystic fibrosis, mitochondrial defects, etc).') for at least 6 months.
3. Insulin pump user for at least 3 months.
4. Treated with rapid or ultra-rapid acting insulin analogue.
5. Subject/carer is willing to perform at least two fingerprick blood glucose measurements per day.
6. Screening HbA1c ≤11% (97 mmol/mol) based on analysis from local laboratory.

7. Able to wear glucose sensor and closed-loop system 24/7.
8. The subject/carer is willing to follow study specific instructions.
9. The subject/carer is willing to upload pump and CGM data at regular intervals.

### Exclusion criteria
1. Physical or psychological disease likely to interfere with the normal conduct of the study.
2. Untreated coeliac disease or thyroid disease.
3. Current treatment with drugs known to interfere with glucose metabolism.
4. Use of closed-loop insulin delivery within the past 2 months.
5. Known or suspected allergy to insulin.
6. Carer's lack of reliable telephone facility for contact.
7. Parent/guardian's severe visual or hearing impairment.
8. Medically documented allergy towards the adhesive (glue) of plasters or subject is unable to tolerate tape adhesive in the area of sensor placement.
9. Serious skin diseases located at places of the body corresponding with sensor insertion sites.
10. Sickle cell disease, haemoglobinopathy, has received red blood cell transfusion or erythropoietin within 3 months prior to time of screening.
11. Plan to receive red blood cell transfusion or erythropoietin over the course of study participation.
12. Subject/carer not proficient in English (UK, Germany, Austria, Luxembourg), German (Germany, Austria, Luxembourg) or French (Luxembourg).
13. Additional national exclusion criteria may apply.

### Study schedule
The study will consist of up to 9 visits and 11 telephone/email contacts (tables 1 and 2). The maximum study duration is 11 months. At the recruitment visit, written informed consent will be obtained from parents/guardians and a screening HbA1c will be taken. Validated surveys assessing participants' and families' quality of life, psychosocial function, diabetes management and response to their current treatment will be completed. Interviews with parents/guardians will gather feedback on their current treatment, the clinical trial and quality of life changes.

### Pre-randomisation training and run-in
All participants and their parents/guardians will be trained on CGM device and study insulin pump. Competency on the use of devices will be assessed, followed by 2–4 weeks run-in period. To assess compliance and ability of the parents/guardians to use the study devices safely, at least 8 days of CGM data must be recorded and safe use of the study insulin pump demonstrated during the last 14 days of the run-in period. CGM data will be used to assess baseline glucose control and may be used for treatment optimisation.

**Table 1** Schedule of study visits when closed-loop intervention precedes sensor-augmented pump therapy

| | Visit/contact | Description | Start relative to previous/next Visit/activity | Duration | Comment |
|---|---|---|---|---|---|
| **Consent and training** | **Visit 1** | Recruitment visit: Consent/assent, HbA1c questionnaires | – | 1–2 hours | |
| | **Visit 2** | CGM start: CGM training, initiation of CGM, competency assessment | Within 2 weeks of Visit 1 (may coincide with Visit 1) | 2–3 hours | May be repeated if competency not achieved |
| | **Visit 3** | Pump start: insulin pump training, study pump initiation, competency assessment | Within 1 week of Visit 2 (may coincide with Visit 2) | 2–4 hours | May be repeated if competency not achieved |
| **Run-in 2–4 weeks** | Contact 1 | Review pump settings and CGM data; adjustment of treatment | After 1 week of Visit 3 (±3 days) | 30 min | |
| | **Visit 4*** | End of run-in, adjustment of treatment, treatment adherence assessment | After 2–4 weeks of Visit 3 (minimum of 2 weeks) | 1 hour | Run-in and Visit four may be repeated if non-compliant |
| | **Randomisation** | | Immediately after Visit 4 | | If compliant with pump & CGM use |
| **CL Intervention 16 weeks** | **Visit 5** | CL initiation: training (CL), competency assessment | Within 1 week of Visit 4 | 1–2 hours | |
| | Contact 2 | Follow-up after CL start | Within 24–48 hours after Visit 5 | 30 min | |
| | Contact 3 | Follow-up, review use of study devices | After 1 week of Visit 5 (±3 days) | 30 min | |
| | Contact 4 | Follow-up, review use of study devices | After 4 weeks of Visit 5 (±2 weeks) | 30 min | |
| | Contact 5 | Follow-up, review use of study devices | After 8 weeks of Visit 5 (±2 weeks) | 30 min | |
| | Contact 6 | Follow-up, review use of study devices | After 12 weeks of Visit 5 (±2 weeks) | 30 min | |
| | **Visit 6** | End of first study arm: device download, HbA1c, questionnaires, interview | After 16 weeks of Visit 5 (±2 weeks) | <1 hour | |
| **Washout period (1–4 weeks)** | | | | | |
| **Sensor-augmented pump therapy 16 weeks** | **Visit 7*** | Sensor-augmented pump therapy initiation: refresher training | After 1–4 weeks of Visit 6 (minimum of 1 week) | 1–2 hours | |
| | Contact 7 | Follow-up after Sensor-augmented pump therapy start | within 24 to 48 hours after Visit 7 | 30 min | |
| | Contact 8 | Follow-up, review use of study devices | After 1 week of Visit 7 (±3 days) | 30 min | |
| | Contact 9 | Follow-up, review use of study devices | After 4 weeks of Visit 7 (±2 weeks) | 30 min | |
| | Contact 10 | Follow-up, review use of study devices | After 8 weeks of Visit 7 (±2 weeks) | 30 min | |
| | Contact 11 | Follow-up, review use of study devices | After 12 weeks of Visit 7 (±2 weeks) | 30 min | |
| | **Visit 8 (caregiver only) (UK and Luxembourg only)** | Sleep assessment | After 12 weeks of Visit 7 (±2 weeks) (Visit 8 may coincide with Contact 11) | 30 min | |
| | **Visit 9** | End of study, return and download of devices, HbA1c, questionnaires, interview, resume standard pump therapy | After 16 weeks of Visit 7 (±2 weeks) | <1 hour | |

*Could be done via phone/email.
CGM, continuous glucose monitoring device; CL, closed-loop system; HbA1c, glycated haemoglobin.

**Table 2** Schedule of study visits when sensor-augmented pump therapy precedes closed-loop intervention

| | Visit/contact | Description | Start relative to previous/ next Visit/activity | Duration | Comment |
|---|---|---|---|---|---|
| **Consent and training** | **Visit 1** | Recruitment visit: Consent/assent, HbA1c, questionnaires | – | 1–2 hours | |
| | **Visit 2** | CGM start: CGM training, initiation of CGM, competency assessment | Within 2 weeks of Visit 1 (may coincide with Visit 1) | 2–3 hours | May be repeated if competency not achieved |
| | **Visit 3** | Pump start: insulin pump training, study pump initiation, competency assessment | Within 1 week of Visit 2 (may coincide with Visit 2) | 2–4 hours | May be repeated if competency not achieved |
| **Run-in 2–4 weeks** | Contact 1 | Review pump settings and CGM data; adjustment of treatment | After 1 week of Visit 3 (±3 days) | 30 min | |
| | **Visit 4*** | End of run-in, adjustment of treatment, treatment adherence assessment | After 2–4 weeks of Visit 3 (minimum of 2 weeks) | 1 hour | Run-in and Visit four may be repeated if non-compliant |
| | **Randomisation** | | Immediately after Visit 4 | | If compliant with pump and CGM use |
| **Sensor-augmented pump therapy 16 weeks** | **Visit 5** | Sensor-augmented pump therapy initiation: refresher training | Within 1 week of Visit 4 | 1–2 hours | |
| | Contact 2 | Follow-up after sensor-augmented pump therapy start | Within 24–48 hours after Visit 5 | 30 min | |
| | Contact 3 | Follow-up, review use of study devices | After 1 week of Visit 5 (±3 days) | 30 min | |
| | Contact 4 | Follow-up, review use of study devices | After 4 weeks of Visit 5 (±2 weeks) | 30 min | |
| | Contact 5 | Follow-up, review use of study devices | After 8 weeks of Visit 5 (±2 weeks) | 30 min | |
| | Contact 6 | Follow-up, review use of study devices | After 12 weeks of Visit 5 (±2 weeks) | 30 min | |
| | **Visit 6** | End of first study arm: device download, HbA1c, questionnaires, interview | After 16 weeks of Visit 5 (±2 weeks) | <1 hour | |
| | **Washout period (1–4 weeks)** | | | | |
| **CL Intervention 16 weeks** | **Visit 7*** | CL initiation: training (CL), competency assessment | After 1–4 weeks of Visit 6 (minimum of 1 week) | 1–2 hours | |
| | Contact 7 | Follow-up after CL start | within 24–48 hours after Visit 7 | 30 min | |
| | Contact 8 | Follow-up, review use of study devices | After 1 week of Visit 7 (±3 days) | 30 min | |
| | Contact 9 | Follow-up, review use of study devices | After 4 weeks of Visit 7 (±2 weeks) | 30 min | |
| | Contact 10 | Follow-up, review use of study devices | After 8 weeks of Visit 7 (±2 weeks) | 30 min | |
| | Contact 11 | Follow-up, review use of study devices | After 12 weeks of Visit 7 (±2 weeks) | 30 min | |
| | **Visit 8 (caregiver only)** (UK and Luxembourg only) | Sleep assessment | After 12 weeks of Visit 7 (±2 weeks) (Visit 8 may coincide with Contact 11) | 30 min | |
| | **Visit 9** | End of study, return and download of devices, HbA1c, questionnaires, interview, resume standard pump therapy | After 16 weeks of visit 7 (±2 weeks) | <1 hour | |

*Could be done via phone/email.
CGM, continuous glucose monitoring device; CL, closed-loop system; HbA1c, glycated haemoglobin.

## Randomisation

Eligible subjects will be randomised after run-in using remote central randomisation software to the initial use of the hybrid closed-loop system or to sensor-augmented pump therapy for 16 weeks, with 1–4 weeks washout period before crossing over to the other study arm. Randomisation will be stratified by site and the randomisation ratio will be 1:1 within each stratum. The randomisation list created by the study statistician is encrypted.

## Post-randomisation training
### Automated closed-loop insulin delivery (intervention arm)

Participants randomised to the closed-loop arm and their parents/guardians will receive a training session by the research team covering the use of the closed-loop system prior to starting closed-loop insulin delivery. Competency on the use of closed-loop system will be assessed. Participants will use the hybrid closed-loop system for 16 weeks at home.

### Sensor-augmented pump therapy (control arm)

Participants in the sensor-augmented pump therapy arm and their parent/guardians will receive refresher training on key aspects of insulin pump therapy and CGM use. CamAPS FX app does not implement a low glucose suspend or predictive low glucose suspend function in open loop. Participants will use sensor-augmented pump therapy for 16 weeks at home.

## Assessments at 4 and 8 months

A blood sample for HbA1c measurement will be taken at the end of each study arm. Validated surveys evaluating the impact of devices employed on quality of life, psychosocial function, diabetes management and treatment satisfaction will be completed. Parents/guardians will be invited to be interviewed to gather feedback on their current treatment, the clinical trial and quality of life changes.

Participants will have 1–4 weeks washout period before commencing the second study arm with training as described above.

In a subset of clinical sites, parents/guardians will be invited to participate in a parallel 7-day sleep substudy prior to the final study visit. All parents/guardians will wear an Actiwatch and keep a sleep diary. Questionnaires assessing sleep quality will be completed by the parents/guardians. At the final visit, participants will resume usual care using their prestudy insulin pump.

## Contacts during home study period

Parents/guardians will be contacted 24 hours after starting each study arm to ensure there are no concerns regarding study devices. In between study visits, parents/guardians will be contacted by the study team by email/telephone once monthly to record any adverse events, device deficiencies, changes in insulin settings, other medical conditions and/or medications.

Routine clinical care will be provided by the local paediatric diabetes team as per usual care. Throughout the trial, parents/guardians and/or the clinical team are free to adjust insulin therapy as per usual clinical practice, but no active treatment optimisation or remote monitoring will be undertaken by the study team.

In case of any problems related to the technical device or diabetes management parents/guardians will be able to contact a 24-hour telephone helpline to the local research team. The local research team will have access to a central helpline for technical issues.

## Withdrawal criteria

A participant/guardian may terminate participation in the study at any time without giving a reason and without any personal disadvantage. An investigator can stop the participation of a subject after consideration of the benefit/risk ratio. Possible reasons are:
► Serious adverse events.
► Significant protocol violation or non-compliance.
► Decision by the investigator, or the sponsor, that termination is in the subject's best medical interest.
► Allergic reaction to insulin.

## STUDY PROCEDURES
### Blood samples

A blood sample for the measurement of HbA1c levels will be taken at three time points: baseline and at the end of each study arm. HbA1c will be measured locally using International Federation of Clinical Chemistry and Laboratory Medicine aligned methods. All HbA1c testing must follow National Glycohaemoglobin Standardisation Programme standards.

## Psychosocial assessments
### Questionnaires

Parents/guardians will be invited to complete a series of validated questionnaires on health-related quality of life at baseline and at the end of each study arm. All questionnaires are available in English, French and German. At the end of the closed-loop intervention arm, parents'/guardians' experience using closed-loop will be documented using the parent closed-loop experience questionnaire.

### Measures of sleep quality

Quality, duration and fragmentation of sleep will be assessed subjectively using the Pittsburgh Sleep Quality Index (PSQI), the Children's Sleep Habit Questionnaire (CSHQ) and the Epworth sleepiness scale (ESS). Parents/guardians will wear an Actiwatch (Philips Respironics, Bend, Oregon, USA) to provide objective measures of sleep and wakefulness based on motor activity and complete a sleep diary. These measures will be evaluated over 7 days prior to the end of the study in both arms.

### Qualitative interviews

A subset of approximately 30 parents/guardians will be interviewed at the end of each study arm with representation from each study country and participant age-group. Parents/guardians will be asked to opt-in to the interview study at recruitment.

The interviews will explore parents'/guardians' experiences of using closed-loop insulin delivery (as compared with sensor-augmented pump therapy) to manage their child's diabetes. The same parents/guardians will be interviewed after completing each study arm to look at whether, in what ways and why, use of a closed-loop system (as compared with sensor-augmented pump therapy) has impacted on their diabetes management practices, their worries and concerns about hyperglycaemia and hypoglycaemia, and their work and family lives. Interviews will explore participants' likes and dislikes of using the closed-loop system and how the technology might be improved for future use. Interviews will take place at a time chosen by the participant and carried out by telephone. Participants will be interviewed in English or German by an experienced qualitative researcher fluent in both languages. All interviews will be transcribed in full for in-depth analysis; with interviews undertaken in German translated into English.

## HEALTH ECONOMICS

Health economics analysis will be performed contrasting closed-loop and sensor-augmented pump therapy using a health economic simulation model: the IQVIA Core Diabetes Model (CDM; QVIA, Basel, Switzerland). The CDM is a validated non-product-specific policy analysis tool for cost-effectiveness analysis in T1D. Long-term outcomes derived from the simulation will include total direct costs, life expectancy, quality-adjusted life expectancy and time to onset of complications. Incremental costs versus incremental effectiveness (quality-adjusted life years (QALYs)) for closed-loop versus sensor augmented pump therapy will be compared. Baseline characteristics of the simulation cohort will come from the study. Treatment effects will be based on the study findings comparing 16 weeks of closed-loop with 16 weeks of sensor-augmented pump therapy. Country-specific direct costs will be sourced from published literature and where necessary inflated to the current year costs. For treatment costs, only the incremental costs between the two interventions will be considered.

## PATIENT AND PUBLIC INVOLVEMENT

The research question and study endpoints are based on feedback from participants of previous studies and in line with prioritisation by stakeholders. The study design and assessment of burden of the intervention were reviewed by focus groups. Results will be disseminated to participants and the general public through social media and made available on the sponsor's website.

## STATISTICAL ANALYSIS

All analyses will be conducted on an intention-to-treat basis. Data from all randomised participants with/without protocol violation including drop-outs and withdrawals will be included in the analysis. The statistical analysis plan can be found in online supplemental information.

### Primary endpoint analysis

The primary analysis will evaluate the between group difference in time spent in the target glucose range from 3.9 to 10.0 mmol/L based on CGM glucose levels during the 16-week study periods. Mean±SD or summary statistics appropriate to the distribution will be reported for the primary endpoint over the 16-week period by treatment intervention. The treatment interventions will be compared using a linear mixed model adjusting for period as a fixed effect and site as a random effect. The model will account for correlated data from the same subject. A 95% CI will be reported for the difference between the interventions based on the linear mixed model. The primary analysis will be a single statistical comparison of a single outcome measure.

### Key secondary endpoints

For the following key endpoints, the familywise type I error rate will be controlled at two-sided α=0.05. A gatekeeping strategy will be used, where the primary endpoint will be tested first, if passing the significance testing, other key endpoints will be tested in the order listed below using the fixed-sequence method at α=0.05.
- ▶ Time spent in target glucose range (3.9 to 10.0 mmol/L).
- ▶ Time spent above target glucose (10.0 mmol/L).
- ▶ HbA1c.
- ▶ Average of glucose levels.
- ▶ Time spent below target glucose (3.9 mmol/L).

### Secondary efficacy endpoints

The following endpoints will be assessed:
CGM derived indices:
- ▶ SD and coefficient of variation of glucose levels.
- ▶ Time with glucose levels <3.0 mmol/L.
- ▶ Time with glucose levels in significant hyperglycaemia (glucose levels>16.7 mmol/L).
- ▶ AUC of glucose below 3.5 mmol/L.
Insulin dose and other endpoints:
- ▶ Body mass index (BMI) SD score.
- ▶ Total, basal and bolus insulin dose.

For all secondary endpoints, summary statistics appropriate to the distribution will be tabulated by treatment group. Analysis of all secondary CGM and insulin endpoints will parallel the primary analysis. For these exploratory analyses, the Benjamini-Hochberg false discovery rate will be used to account for multiple comparisons. The models comparing BMI SD score and HbA1c will also adjust for baseline value at the start of each period. A ranked normal score transformation will be applied to all highly skewed secondary endpoints.

Summary statistics for the following outcome metrics will also be tabulated separately for daytime (defined as 8:00 hours to less than 24:00 hours) and night-time

(defined as 00:00 hours to less than 8:00 hours) over the 16-week period:

► Percent time with glucose levels spent in the target range (3.9 to 10.0 mmol/L).
► Mean of glucose levels.
► SD of glucose levels.
► Per cent time with glucose levels below 3.9 mmol/L.
► Total insulin dose.

### Safety analysis
The following events will be recorded and compared between study arms:

► Number of subjects with any diabetic ketoacidosis events.
► Number of episodes of diabetic ketoacidosis events per subject and incidence rate per 100 person-years.
► Number of subjects with any severe hypoglycaemia events.
► Number of episodes of severe hypoglycaemia events per subject and incidence rate per 100 person-years.
► Number of any other adverse events per subject.
► Number of any other serious adverse events per subject.

All safety outcomes will be tabulated for all subjects, including dropouts and withdrawals, regardless of whether CGM data are available and irrespective of whether closed-loop was operational. All adverse events will be listed for the entire study duration, including run-in and washout period. For diabetic ketoacidosis and severe hypoglycaemia events, the event rates will be compared using a repeated measures Poisson regression model adjusting for study arm. Binary safety outcomes will also be compared using a repeated measures logistic regression model adjusting for study arm.

### Utility evaluation
The following system use will be tabulated:

► Percentage of time CGM is used for each 16-week treatment period.
► Percentage of time when closed-loop system use is functioning during the closed-loop period

### Psychosocial evaluation
#### Questionnaires
For each questionnaire (and their corresponding subscales), total scores will be calculated and reported at each time point. They will also be compared between treatment periods using the same model described for the primary endpoint. The distribution of responses for each individual question at each time point will be reported in a separate table.

#### Sleep quality
The PSQI, CSHQ, ESS and actigraph data will be used to calculate mean total sleep quality score, sleep duration (sum of all epochs scored as sleep during the time in bed) and variability across nights, time in bed, sleep disturbance (including wake after sleep onset and number of awakenings), latency, efficiency, quality and daytime

dysfunction. Sleep data will be averaged across nights in each parent/guardian for the parallel study period.

### Qualitative interviews
To maximise rigour at least two experienced qualitative researchers will be involved in data analysis using a thematic approach. NVivo V.11, a qualitative data analysis software, will be used to facilitate data management.

### Health economics
For each simulation, a simulated cohort of 1000 patients will be run through the model 1000 times using first-order Monte Carlo simulation. Long-term outcomes derived from simulation will include total direct costs, life expectancy, quality-adjusted life expectancy and time to onset of complications. Future costs and clinical benefits will be discounted based on each country's recommendations. Incremental costs versus incremental effectiveness (QALY) of the two therapies will be compared.

Additional exploratory health economic analysis will be conducted on other endpoints such as, but not limited to, sleep disorders and indirect costs for society.

### Interim analysis
No formal interim analyses or stopping guidelines are planned for this study.

### Power calculation
Data from the SENCE study (Strategies to Enhance New CGM Use in Early Childhood study, NCT02912728) were used to estimate the SD of the primary endpoint. Based on data from this study, 65 subjects are required to attain 90% power to detect a difference if a treatment effect of 5%, an SD of 10.3% for an individual measurement, and a correlation of 0.3 between periods are assumed. Adding an additional 10% to this sample size to account for drop-outs, results in a final sample size of 72 randomised participants.

## STUDY MANAGEMENT
Composition of study management groups is shown in online supplemental appendix.

### Data and safety monitoring board
An independent data and safety monitoring board (DSMB) will be informed of all serious adverse events and unanticipated adverse device effects that occur during the study. The DSMB will review compiled adverse event data at periodic intervals and will report to the trial steering committee (TSC) any safety concerns and recommendations for suspension or early termination of the trial.

### Study sponsor
The study sponsors are the Cambridge University Hospitals NHS Foundation Trust and the University of Cambridge.

### Trial steering committee

The TSC will provide the overall supervision of the clinical trial. The TSC will comprise an independent chairperson, the chief investigator and leaders of work packages (WP2 (pilot study), WP3 (main study) and WP4 (data management)) of the KidsAP consortium. The TSC will monitor clinical trial progress and advise on scientific credibility. The TSC will consider and act, as appropriate, on the recommendations of the DSMB. The TSC will report its decisions to the Ethical Board and the General Assembly of the KidsAP Consortium.

### Trial management group

The trial management group will meet weekly and will be responsible for day-to-day management of the trial.

### Data management and monitoring

The study coordinators will be responsible for maintaining quality assurance and quality control systems to ensure that the trial is conducted and data are generated, documented and reported in compliance with the protocol, good clinical practice and regulatory requirements.

Confidentiality of participant data shall be observed at all times. Personal details for each participant taking part with a link to a unique identification number will be held locally on a study screening log in the trial site file at each study site. These details will not be revealed at any other stage during the study, and all results will remain anonymous.

Case report forms will be used for recording anonymised study data and will be completed in accordance with Good Clinical Practice and ISO 15197: 2013 guidelines.

### Indemnity

Indemnity for any harm arising from the conduct of research will be provided according to local arrangements in respective centres.

► Cambridge, UK-National Health Service (NHS) indemnity cover will apply for any claims arising from management and conduct of research. Any liability arising from study design will be covered by the clinical trial insurance policy organised by the University of Cambridge.
► Graz, Innsbruck, Vienna, Austria—Subjects will be insured according to Medical Device Law § 47 (StF: BGBl. Nr. 657/1996, BGBl. I Nr. 143/2009)
► Leipzig, Germany—Subjects will be insured according to the German Medical Device Law (MPG). Any liability arising from the study will be covered by the clinical trial insurance policy organised by the University of Leipzig.
► Luxembourg—Centre Hospitalier de Luxembourg indemnity will apply for any claims arising from the management and conduct of research. Any liability arising from the study design will be covered by the clinical trial insurance policy organised by the Centre Hospitalier de Luxembourg (Amlin Corporate Insurance NV).

## ETHICS AND DISSEMINATION

This study has received approvals from the Cambridge East Research Ethics Committee in the UK (18/EE/0290), the Ethics Committee of the University of Innsbruck in Austria (EK Nr 1238/201), the Ethics Committee of the University of Vienna in Austria (EK Nr 1979/2018), the Ethics Committee of the University of Graz in Austria (EK Nr 31–077 ex 18/19), the Ethics Committee of the Medical Faculty of the University of Leipzig in Germany (029/19-ff), the Comité National d'Ethique de Recherche in Luxembourg (201812/01) and has undergone review and approval by regulatory authorities in the UK by the Medicines and Healthcare products Regulatory Agency, in Austria by the Austrian Medicines and Medical Devices Agency, in Germany by Bundesinstitut für Arzneimittel und Medizinprodukte, and in Luxembourg by the Ministry of Health.

All parents/guardians will be provided with oral and written information about the trial and study procedures before obtaining written informed consent. Standard operating procedures for monitoring and reporting of all adverse events and adverse device effects will be in place including serious adverse events, serious adverse device effects and specific adverse events such as severe hypoglycaemia and significant hyperglycaemia with ketosis. Any substantial amendments to the protocol and other documents shall be notified to and approved by the independent REC and the regulatory authorities, prior to implementation as per nationally agreed guidelines. Screening and recruitment commenced in May 2019, and the study is expected to be completed by June 2021. Study results will be disseminated by peer-reviewed publications and conference presentations.

**Author affiliations**
[1]Wellcome Trust-MRC Institute of Metabolic Science, University of Cambridge, University of Cambridge School of Clinical Medicine, Cambridge, UK
[2]Department of Paediatrics, University of Cambridge School of Clinical Medicine, Cambridge, UK
[3]DECCP, Clinique Pédiatrique, Centre Hospitalier de Luxembourg, Luxembourg City, Luxembourg
[4]Leeds Children's Hospital, Leeds, UK
[5]Department of Pediatric and Adolescent Medicine, Medical University of Graz, Graz, Steiermark, Austria
[6]Department of Internal Medicine, Division of Endocrinology and Diabetology, Medical University of Graz, Graz, Steiermark, Austria
[7]Department of Pediatrics I, Medical University of Innsbruck, Innsbruck, Tirol, Austria
[8]Hospital for Children and Adolescents, University of Leipzig Faculty of Medicine, Leipzig, Sachsen, Germany
[9]Department of Pediatrics and Adolescent Medicine, Medical University of Vienna, Wien, Austria
[10]Endocrinology, Stanford University School of Medicine, Stanford, California, USA
[11]The University of Edinburgh Usher Institute of Population Health Sciences and Informatics, Edinburgh, UK
[12]Vyoo Agency Sarl, Lyon, France
[13]Jaeb Centre for Health Research, Tampa, Florida, USA

**Acknowledgements**  Jasdip Mangat supported development and validation of the closed-loop system. Cambridge Clinical Trials Unit, Nicole Ashcroft, Josephine Hayes and Matthew Haydock (Institute of Metabolic Science, University of Cambridge) provide administrative support. Artificial pancreas focus group contributors provided feedback on the study design.

**Collaborators** The following are members of the KidsAP Consortium (CI - chief investigator, PI - principal investigator, I - investigator, SC - study coordinator, SN - study nurses, RA - research assistant, AM - administrative manager). Cambridge, UK: Roman Hovorka (CI and project lead), Carlo L Acerini (PI), Ajay Thankamony (PI), Charlotte K Boughton (I), Klemen Dovc (I), Julia Fuchs (I), Gianluca Musolino (I), Malgorzata E Wilinska (I), Janet M Allen (SN), Nicole Ashcroft (SC), Matthew Haydock (SC). Luxembourg: Carine de Beaufort (PI), Ulrike Schierloh (I), Muriel Fichelle (SN), Dominique Schaeffer (SN). Leeds, UK: Fiona Campbell (PI), James Yong (I), Emily Metcalfe (SN), Saima Waheed (SN), Joseph Tulip (RA). Graz, Austria: Elke Froehlich-Reiterer (PI), Maria Fritsch (I), Hildegard Jasser-Nitsche (I), Julia K Mader (I), Kerstin Faninger (SN). Innsbruck, Austria: Sabine E Hofer (PI), Daniela Abt (SN), Anita Malik (AM), Barbara Lanthaler (AM), Matthias Wenzel (AM). Leipzig, Germany: Thomas M Kapellen (PI), Heike Bartelt (I), Alena Thiele (RA). Vienna, Austria: Birgit Rami-Merhar (PI), Gabriele Berger (I), Nicole Blauensteiner (I), Renata Gellai (I), Katrin Nagl (I), Martin Tauschmann (I), Sarah Cvach (SN), Sonja Katzenbeisser-Pawlik (SN). Stanford University, CA, USA: Korey K Hood (I). Usher Institute, Edinburgh, UK: Barbara Kimbell (I), Julia Lawton (I). VYOO Agency, Lyon, France: Stephane Roze (I). Jaeb Centre for Health Research, Tampa, FL, USA: Nathan Cohen (RA), Craig Kollman (RA), Judy Sibayan (RA).

**Contributors** RH, CdB, FC, EF-R, JKM, SEH, TMK, BR-M, MT, JY, CKB, MEW, JMA, NC and AT codesigned the study. CdB, FC, EF-R, SEH, TMK, BR-M and AT are the lead clinical investigators. KH, BK and JL will conduct the psychosocial assessments. JS supported study setup. SR will conduct the cost utility analysis. RH designed and implemented the glucose controller. JF and RH wrote the manuscript. All authors critically reviewed the report. No writing assistance was provided.

**Funding** This work was supported by the European Commission within the Horizon 2020 Framework Programme under the grant agreement number 731560 and JDRF (grant number N/A). Additional support for the artificial pancreas work from National Institute for Health Research Cambridge Biomedical Research Centre (grant number N/A), and Wellcome Strategic Award (100574/Z/12/Z). The views expressed are those of the author(s) and not necessarily those of the NIHR or the Department of Health and Social Care. Dexcom is supplying discounted CGM devices (grant number N/A).

**Competing interests** SEH declares speaker honoraria from Eli Lilly, Pfizer and Sanofi. EF-R reports having received speaker honoraria from Minimed Medtronic and Eli Lilly, serving on advisory boards for Eli Lilly. TMK has received speaker honoraria from Minimed Medtronic, Roche and Eli Lilly and consulted Sanofi-Aventis. CdB has received speaker honoraria from Minimed Medtronic, and is member of their European Psychology Advisory Board. BR-M reports having received speaker honoraria from Abbott, Minimed Medtronic, Eli Lilly, Roche, Menarini and Novo Nordisk, serving on advisory boards for Eli Lilly. FC has received speaking fees from Abbott, Medtronic and Eli Lilly. RH reports having received speaker honoraria from Eli Lilly and Novo Nordisk, serving on advisory panel for Eli Lilly and Novo Nordisk; receiving license fees from BBraun and Medtronic; having served as a consultant to BBraun, patents and patent applications related to closed-loop insulin delivery, and director at CamDiab. JKM is shareholder of Decide Clinical Software, a member in the advisory board of Boehringer Ingelheim, Eli Lilly, Medtronic, Prediktor A/S, Roche Diabetes Care, Sanofi-Aventis and received speaker honoraria from Abbott Diabetes Care, Astra Zeneca, Dexcom, Eli Lilly, NovoNordisk A/S, Roche Diabetes Care, Servier and Takeda.

**Patient consent for publication** Not required.

**Provenance and peer review** Not commissioned; externally peer reviewed.

**ORCID iDs**
Julia Fuchs http://orcid.org/0000-0002-4497-0979
Charlotte K Boughton http://orcid.org/0000-0003-3272-9544

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
