## [Reviewer comments · BMJ Open]

ARTICLE DETAILS

TITLE (PROVISIONAL)	Assessing the efficacy, safety and utility of closed-loop insulin delivery compared to sensor-augmented pump therapy in very young children with type 1 diabetes (KidsAP02 study): an open-label, multi-centre, multi-national, randomised crossover study protocol
AUTHORS	Fuchs, Julia; Allen, Janet; Boughton, Charlotte; Wilinska, Malgorzata; Thankamony, Ajay; de Beaufort, Carine; Campbell, Fiona; Yong, James; Froehlich-Reiterer, Elke; Mader, Julia; Hofer, Sabine; Kapellen, Thomas; Rami-Merhar, Birgit; Tauschmann, Martin; Hood, Korey; Kimbell, Barbara; Lawton, Julia; Roze, Stephane; Sibayan, Judy; Cohen, Nathan; Hovorka, Roman

VERSION 1 – REVIEW

REVIEWER	Dr Monika Reddy Imperial College London United Kingdom
REVIEW RETURNED	29-Oct-2020

GENERAL COMMENTS	This manuscript outlines the protocol for a clinical study comparing a closed-loop insulin delivery system (intervention) with sensor-augmented pump therapy (control) in 72 very young children with type 1 diabetes. This is an important study and the study protocol is well-thought through. The study is led by a team of investigators with a longstanding track record in closed-loop insulin delivery research. The manuscript is well-written and clear. I look forward to seeing the results of this study. I have a few minor comments and suggestions: 1. It would be helpful if the authors could justify or explain why predictive low glucose suspend (PLGS) was not used in the control group?2. Include 780G as a commercially available Hybrid AP System in the introduction.3. If possible please include some further details of the health economic analysis plan in the main manuscript.4. Please clarify if there will be any remote monitoring of participants' glucose and other aspects of algorithm during the study.5. Please clarify whether the investigators will be enabled to change any of the settings (insulin: carb ratio, correction factor, basal rates or the aggressiveness of the AP algorithm) once the participant has entered into the intervention or control arm i.e. for safety reasons?
---

VERSION 1 – AUTHOR RESPONSE

Reviewer: 1

Dr. Monika Reddy, Imperial College London

Competing interests of Reviewer: None declared

Comments to the Author:

This manuscript outlines the protocol for a clinical study comparing a closed-loop insulin delivery system (intervention) with sensor-augmented pump therapy (control) in 72 very young children with type 1 diabetes. This is an important study and the study protocol is well-thought through. The study is led by a team of investigators with a longstanding track record in closed-loop insulin delivery research. The manuscript is well-written and clear. I look forward to seeing the results of this study.

RESPONSE: We would like to thank the Reviewer for this positive feedback.

I have a few minor comments and suggestions:

1. It would be helpful if the authors could justify or explain why predictive low glucose suspend (PLGS) was not used in the control group?

RESPONSE: Thank you for this important comment. The CamAPS FX app does not implement a LGS or PLGS function as its intended use is for closed-loop insulin delivery. There is no Dexcom-compatible PLGS system currently approved for children <6 years old, so we were not able to use another pump with PLGS feature. The Medtronic 640G PLGS system is only compatible with Enlite sensors and switching pump and sensor in each arm would introduce considerable device bias.

To clarify this in the paper we have added the following sentence on page 11, paragraph 4: "CamAPS FX app does not implement a predictive low glucose suspend function."

2. Include 780G as a commercially available Hybrid AP System in the introduction.

RESPONSE: We thank the Reviewer for this suggestion and have included the 780G system in paragraph 3 of the introduction.

3. If possible please include some further details of the health economic analysis plan in the main manuscript.

RESPONSE: Thank you for this suggestion. In order to provide more detail we have included the following on page 14, paragraph 2: "Long-term outcomes derived from the simulation will include total direct costs, life expectancy, quality-adjusted life expectancy and time to onset of complications. Incremental costs versus incremental effectiveness (quality-adjusted life years [QALYs]) for closed-loop vs sensor augmented pump therapy will be compared."

4. Please clarify if there will be any remote monitoring of participants' glucose and other aspects of algorithm during the study.

RESPONSE: We thank the Reviewer for her input. There will be no remote monitoring of glucose levels or other aspects of the algorithm by study staff outside of study contacts and visits, but parents/guardians have access to SMS and Diasend remote monitoring. To clarify this is in the manuscript we have amended the following sentence on page 12, paragraph 5: "Throughout the trial, parents/guardians and/or the clinical team are free to adjust insulin therapy as per usual clinical practice, but no active treatment optimisation or remote monitoring will be undertaken by the study team."

5. Please clarify whether the investigators will be enabled to change any of the settings (insulin: carb ratio, correction factor, basal rates or the aggressiveness of the AP algorithm) once the participant has entered into the intervention or control arm i.e. for safety reasons?

RESPONSE: We thank the Reviewer for this comment. We refer to our response to comment 4 with regards to study staff not undertaking active treatment optimisation during the trial. However in terms of safety, parents/guardians will be able to contact a 24-hour telephone helpline to the local research team in case of any problems related to the technical device or diabetes management (manuscript page 12, paragraph 6). The local research team will have access to a central helpline for technical issues. We hope this clarifies the issue.